# Prolactin Rescues Immature B Cells from Apoptosis-Induced BCR-Aggregation through STAT3, Bcl2a1a, Bcl2l2, and Birc5 in Lupus-Prone MRL/lpr Mice

**DOI:** 10.3390/cells10020316

**Published:** 2021-02-04

**Authors:** Rocio Flores-Fernández, Angélica Aponte-López, Mayra C. Suárez-Arriaga, Patricia Gorocica-Rosete, Alberto Pizaña-Venegas, Luis Chávez-Sanchéz, Francico Blanco-Favela, Ezequiel M. Fuentes-Pananá, Adriana K. Chávez-Rueda

**Affiliations:** 1UIM en Inmunologia, Hospital de Pediatría, CMN SIGLO XXI, Instituto Mexicano del Seguro Social, Mexico City 06720, Mexico; chio_tom@hotmail.com (R.F.-F.); luis_chz@hotmail.com (L.C.-S.); fblanco5@hotmail.com (F.B.-F.); 2Unidad de Investigación en Virología y Cáncer, Hospital Infantil de Mexico Federico Gómez, Mexico City 06720, Mexico; angibel06@hotmail.com (A.A.-L.); mostancmayra@hotmail.com (M.C.S.-A.); 3Programa de Doctorado en Ciencias Biomédicas, Universidad Nacional Autónoma de Mexico, Mexico City 04510, Mexico; 4Laboratorio de Biotecnología y Bioinformática Genómica, ENCB, Instituto Politécnico Nacional, Mexico City 11340, Mexico; 5Departamento de Investigación en Bioquímica, Instituto Nacional de Enfermedades Respiratorias “Ismael Cosió Villegas”, Mexico City 14080, Mexico; gorocicap@gmail.com; 6Unidad de Investigación y Bioterio, Instituto Nacional de Enfermedades Respiratorias “Ismael Cosió Villegas”, Mexico City 14080, Mexico; apv38@hotmail.com

**Keywords:** prolactin receptor, prolactin, immature B cells, STAT3, Birc5, Bcl2a1a, systemic lupus erythematosus, MRL/lpr mice

## Abstract

Self-reactive immature B cells are eliminated through apoptosis by tolerance mechanisms, failing to eliminate these cells results in autoimmune diseases. Prolactin is known to rescue immature B cells from B cell receptor engagement-induced apoptosis in lupus-prone mice. The objective of this study was to characterize in vitro prolactin signaling in immature B cells, using sorting, PCR array, RT-PCR, flow cytometry, and chromatin immunoprecipitation. We found that all B cell maturation stages in bone marrow express the prolactin receptor long isoform, in both wild-type and MRL/lpr mice, but its expression increased only in the immature B cells of the latter, particularly at the onset of lupus. In these cells, activation of the prolactin receptor promoted STAT3 phosphorylation and upregulation of the antiapoptotic Bcl2a1a, Bcl2l2, and Birc5 genes. STAT3 binding to the promoter region of these genes was confirmed through chromatin immunoprecipitation. Furthermore, inhibitors of prolactin signaling and STAT3 activation abolished the prolactin rescue of self-engaged MRL/lpr immature B cells. These results support a mechanism in which prolactin participates in the emergence of lupus through the rescue of self-reactive immature B cell clones from central tolerance clonal deletion through the activation of STAT3 and transcriptional regulation of a complex network of genes related to apoptosis resistance.

## 1. Introduction

B cell ontogeny starts in the bone marrow (BM) and occurs through several sequential stages, from pro-B and pre-B to immature B cells. Immature B cells are the first ones expressing surface IgM, exiting the BM to complete maturation in secondary lymph organs [1]. During B cell development, there are several checkpoints that remove self-reactive clones through tolerance mechanisms (receptor editing, clonal deletion, and anergy). In the BM, central tolerance is conducted in immature B cells, removing approximately 85% of them [2,3]. Defects in these tolerance processes have been implicated in the pathogenesis of autoimmune diseases such as systemic lupus erythematosus (SLE) [4].

SLE is a chronic systemic autoimmune disease with an abnormal interplay between innate and adaptive immunity, breach of immune tolerance, production of autoantibodies, and immune complex deposition with multiple organ damage. SLE is considered a multifactorial disease in which immunological, genetic, epigenetic, environmental, and hormonal aspects play an important role in its development. SLE typically affects young women at reproductive age (with a female:male ratio of 9:1); the female predominance has been attributed to the immunostimulatory properties of hormones such as prolactin (PRL) [5,6]. Hyperprolactinemia has been reported in 15–33% of patients with lupus, and PRL levels have shown a direct correlation with the clinical and serological disease activity [7,8,9]. We have previously shown that high levels of PRL exacerbate the disease in a murine experimental model of SLE-like disease; we observed an increase in anti-dsDNA antibodies, autoantibody deposition, and damage to kidneys, which was evidenced through pathogenic proteinuria levels. We observed the expression of the PRL-receptor (extracellular domain) in all maturation stages of B cells, both in control mice (C57BL/6) and mice that developed SLE (MRL and MRL/lpr). However, the level of receptor expression through the B cell maturation stages was different between the control and disease models [10,11]. Furthermore, we found that PRL can rescue immature B cells from apoptosis after BCR (B cell receptor) cross-linking in the MRL/lpr strain and in the WEHI-231 cell line [12].

Different isoforms of the PRL receptor carry identical extracellular but different intracellular domains, which determine the activation of different signaling transduction pathways. One long and three short isoforms have been reported in mice. PRL binding to the long isoform results in a wide range of activation of numerous kinases, including JAK2, STAT, MAPK, PI3K, and AKT [13,14]. In contrast, the short isoforms suppress signaling through the JAK2–STAT pathway due to heterodimer formation with the long isoform [15]. The short isoforms also exhibit long isoform-independent activities upon PRL engagement [16]. Thus, the activated pathway and the resulting PRL effector functions depend on the isoform and cell type in which the PRL-receptor is engaged [15,17,18]. For instance, PRL promotes the proliferation, differentiation, and survival of mammary glands through JAK–STAT5 and PI3K–AKT signaling [19], while it activates STAT1 and STAT5 in endometrial cells [20].

In this study, we determined the isoform of the receptor expressed on immature B cells of wild-type and lupus-prone MRL/lpr mice, and the signaling pathways activated upon PRL receptor engagement to understand the mechanism through which PRL rescues autoimmune cells from apoptosis-induced clonal deletion. We found that BM-maturing B cells expressed only the PRL receptor long isoform. When immature B cells were activated with PRL, it induced the activation of STAT3, which transcriptionally regulated the expression of apoptosis-related genes, mainly *Bcl2l2*, *Bcl2a1a*, *Birc5*, *Gadd45a*, and *Casp8*, thus rescuing immature B cells from apoptosis.

## 2. Materials and Methods

### 2.1. Cell Line

The WEHI-231 murine B lymphoma cell line (ATCC, Manassas, VA, USA) [21] was maintained in RPMI 1640 medium (Hyclone, Logan, UT, USA) supplemented with 10% heat-inactivated fetal bovine serum (FBS, Hyclone), 1% penicillin and streptomycin (Invitrogen, Carlsbad, CA, USA), 1% sodium pyruvate (Hyclone), and 0.1% β-mercaptoethanol (Invitrogen, Carlsbad, CA, USA) at 37 °C under 5% CO_2_.

### 2.2. Mice

Mouse strains were bred in the specific pathogen-free animal facility of the Instituto Nacional de Enfermedades Respiratorias “Ismael Cosio Villegas”. All studies were approved by the Animal Care Committee of the Instituto Nacional de Enfermedades Respiratorias “Ismael Cosio Villegas” and the Hospital de Pediatria, Centro Medico Nacional Siglo XXI, IMSS (protocol number R-2017-785-114); 9–15-week-old mice were used for all experiments, which were performed in accordance with approved guidelines established in Mexico (NOM-062-ZOO-1999) and by the NIH Guide for the Care and Use of Laboratory Animals. MRL/MpJFASlpr (MRL/lpr) mice were purchased from Jackson Laboratory (Bar Harbor, ME, USA) and C57BL/6 mice were purchased from Harlan (Indianapolis, IN, USA).

### 2.3. Prolactin Hormone and Inhibitors

We used murine recombinant PRL (National Hormone and Peptide Program, NIH). PRL receptor activation was ablated with an inhibitor (PRL-inh, G129R), a recombinant analog of the PRL with a single amino acid substitution to create an antagonist of the PRL receptor, which was donated by the manufacturer (Oncolix, Houston, Tex, USA) [22]. Stattic (Cell Signaling Technology, Danvers, MA, USA) is a nonpeptidic selective STAT3 inhibitor. This small molecule prevents the binding of tyrosine-phosphorylated peptide motifs to the STAT3 SH2 domain [23].

### 2.4. Antibodies

The following antibodies were used: anti-B220 MicroBeads (clone RA3-6B2), PE-conjugated anti-B220 (clone RA3-6B2), PE-Cy7-conjugated anti-CD23 (clone B3B4), APC-conjugated anti-IgM (clone 11/41), PE-conjugated anti-AKT (clone REA677), PE-conjugated anti-ERK1/2 (clone REA152); all these were from Miltenyi Biotec (Bergisch Gladbach, Germany). We also used FITC-conjugated anti-CD43 (clone eBioR2/60) and VioBlue-conjugated anti-STAT5 (clone SRBCZX) from eBioscience (San Diego, Cal, USA); PE-conjugated anti-STAT1 (clone 1A5158B) and PE-conjugated anti-STAT3 (clone 13A3-1) from BioLegend (San Diego, Cal, USA); anti-IgM F(ab′)2 antibody (polyclonal), Jackson Immunoresearch (West Grove, PA, USA); and phospho-Stat3 (Tyr705) rabbit IgG antibody (#9131) and normal rabbit IgG (#2729) from Cell Signaling Technology Company.

### 2.5. Purification of B Cells (B220+CD23−) from the Bone Marrow

Nine-week-old mice were euthanized, and BM cells were collected through flushing the femoral shafts with cold RPMI supplemented with 2% FBS and 2 mM EDTA (IBI Scientific, Dubuque, IA, USA). Red blood cells were depleted with a lysis buffer (Sigma-Aldrich, St. Louis, MO, USA) and incubated with anti-CD23 MicroBeads; mature recirculating B cells were removed with the magnetically activated cell-sorting (MACS) system (Miltenyi Biotec) through positive selection using LS columns (Miltenyi Biotec). The negative fraction was recovered and subsequently incubated with anti-B220 MicroBeads (Miltenyi Biotec), so that B220+CD23- BM cells were purified by positive selection.

### 2.6. Analysis of the Signaling Pathways

WEHI-231 cells and B220+CD23– cells from the BM (9-week-old mice) were incubated for 30 min in basal media alone or with the following signaling pathway inhibitors (10 μM): G129R to inhibit the PRL receptor, GSK690693 to inhibit PI3K (Sigma-Aldrich), and Stattic to inhibit STAT3. Subsequently, cells were incubated with PRL (50 ng/mL) for 30 min and fixed with 1× BD Phosflow Lyse/FIx Buffer 5× (BD Biosciences, San Jose, Cal, USA) for 10 min. Cells were permeabilized with Perm Buffer III from BD Phosflow (BD Biosciences) to determine STAT3 phosphorylation and activity in apoptosis rescue assays, and with IC Fixation Buffer (eBioscience) to determine AKT and ERK1/2 phosphorylation at 4 °C for 30 min. Cells were washed with FACS buffer or with Permawash and incubated at 4 °C for 30 min with the antibodies for flow cytometry analysis. Data were acquired using a MACSQuant Analyzer 10 cytometer (Miltenyi Biotec) and analyzed with FlowJo software (Tree Star, Ashland, OR, USA).

### 2.7. Sorting B Cells from BM

Single-cell suspensions of B220+ B cells from BM were incubated with fluorescently labeled antibodies specific for CD43, B220, IgM, and CD23 in a staining buffer (PBS with 0.5% BSA) for 20 min at 4 °C. Further, the cells were incubated with DAPI to select living cells (DAPI−) and washed, then pro-B (B220+CD23-CD43+IgM–), pre-B (B220+CD23–CD43–IgM–), and immature B cells (B220+CD23–CD43–IgM+) were isolated [11]. Cell sorting was performed using a FACS Influx Sorter (BD Biosciences). The purity of sorted cells ranged from 95% to 98%.

### 2.8. RT-PCR for PRL-Receptor Isoforms

To determine the expression of PRL receptor isoforms, real-time PCR was performed using primers synthesized by Integrated DNA Technologies (IDT, Coralville, IA USA): for B-actin (housekeeping control): 5′-GAGGAGGCTCTGGTTCAACA-3′ (left) and 5′-CAGTAAATGCCACGAACGAA-3′ (right). To determine the PRL receptor isoforms, 3 primers were used: common 5′-AAGCCAGACCATGGATACTGGAG-3′ (left), long isoform 5′-AGCAGTTCTTCAGACTTGCCCTT-3′ (right), and short isoform 5′-TTGTATTTGCTTGCAGAGCCAGT-3′ (right). The samples were run in the LightCycler II thermal cycler (Roche, Germany) under the following conditions: 1 cycle of 95 °C for 15 min; 40 cycles of 95 °C for 10 s, 61 °C for 30 s, and 72 °C for 30 s; and 1 cycle of 72 °C for 30 s. The relative expression was analyzed using the 2^–ΔΔCt^ formula. The murine breast cancer cell line EpH4 1424 was used as a positive control for expression of the long and short PRL-receptor isoforms.

### 2.9. PCR Array and RT-PCR

The expression of 84 genes related to apoptosis was evaluated in immature B cells from both C57BL/6 and MRL/lpr mice using the Mouse Apoptosis RT2 Profiler PCR Array (PAMM-012Z, Qiagen, The Netherlands) in a 96-well plate format compatible with the LightCycler 96 thermocycler. Sample processing was performed according to the manufacturer’s instructions using RT^2^ SYBR Green qPCR Mastermix (Qiagen, Spoorstraat KJ Venlo, Netherlands ). The results were analyzed with the Gene Globe Data Analysis Center (Qiagen), which normalized all gene expression data and generated the relative expression value. The expression of some genes was confirmed through RT-PCR using the LightCycler 96 thermocycler. The primers used were synthesized by Integrated DNA Technologies (IDT) and were BCL2L1 5′-GGACCGCGTATCAGAG-3′ (left) and 5′-GCATTGTTCCCGTAGAG-3′ (right), and BIRC5 5′-CCCGATGACAACCCGATA-3ʹ (left) and 5′-CATCTGCTTCTTGACAGTGAGG-3′ (right). The PCR conditions were as follows: 1 cycle of 95 °C for 15 min; 40 cycles of 95 °C for 10 s, 61 °C for 30 s, and 72 °C for 30 s; and 1 cycle of 72 °C for 30 s. The relative expression was analyzed using the 2^–ΔΔCt^ formula.

### 2.10. Analysis of Gene Expression

The transcriptional profile of 84 genes related to apoptosis in immature B cells from 9-week-old mice was analyzed under the following conditions: non-stimulated (control), stimulated with PRL alone, or previously treated with the inhibitors G129R and Stattic, as described in the analysis of signaling pathways. One experiment was performed for each experimental condition. To identify the optimal normalization for gene expression among the set of housekeeping genes included in the array, NormFinder V20 software (Aarhus University, Aarhus, Denmark was used [24]. GAPDH was found to be the most stable gene under all experimental conditions (Appendix A), and thus gene expression was calculated relative to this housekeeping gene using the 2^–ΔΔCt^ method (Appendix A).

After normalization, data analysis was performed in the Gene Globe Data Analysis Center web portal (Qiagen). We first assessed the PRL-altered genes in C57BL/6 and MRL/lpr cells independently, using their respective unstimulated conditions as reference values. Genes with fold change values higher than 2.5 over unstimulated cells were considered to be significantly altered. We then compared the PRL-altered genes in both MRL/lpr and C57BL/6 immature B cells. For this, a nonsupervised hierarchical clustergram and a heat map of gene expression were constructed. We also evaluated the specific genes altered by PRL in MRL/lpr immature B cells by comparing PRL conditions against the inhibitors in a nonsupervised hierarchical clustergram and a heat map of gene expression. To confirm the genes altered directly by PRL stimulation, we performed a supervised analysis, using the unstimulated and the inhibitors cells as control groups. Altered genes shared by different conditions were visualized using Venn diagrams in the Bioinformatics and Evolutionary Genomics portal.

### 2.11. Enrichment of Transcription Factors and Biological Processes

To infer transcription factors related to the differentially expressed genes after PRL stimulation of immature B cells, we used the ChEA3-Chip-X Enrichment Analysis Version 3 (https://amp.pharm.mssm.edu/chea3/). As input, we used the list of upregulated and downregulated genes observed after the analysis of PRL vs. the control and inhibitors in the MRL/lpr immature-B cells, and upregulated and downregulated genes were analyzed independently. The transcription factors with the most significant scores were chosen, and we created a new list that included transcription factors and the upregulated and downregulated genes to build a protein–protein interaction network using STRING v1.4.2 implemented (https://apps.cytoscape.org/apps/stringapp) in Cytoscape v3.6.1 (https://apps.cytoscape.org). The protein–protein interaction network was determined using an interaction score of 0.4, and each interaction was defined by neighborhood, gene fusion, co-occurrence, co-expression, experimental evidence, databases, and text mining. A functional enrichment analysis was conducted using STRING enrichment in STRING v1.4.2 in Cytoscape v3.6.1. The results were represented in the protein–protein interaction network through a donut chart, where each process was represented by a different color. Additionally, we determined the centrality measures closeness and betweenness centrality in Cytoscape. Each parameter was represented in the protein–protein interaction network. The same workflow was utilized to analyze the upregulated and downregulated genes.

### 2.12. Chromatin Immunoprecipitation Assays

WEHI-231 cells and immature B cells from 9-week-old C57BL/6 and MRL/lpr mice were incubated with PRL for 1 h. The EpiTect ChIP OneDay Kit (Qiagen) protocol was used, which includes an antibody anti-RNA Polymerase II as a positive control. A rabbit IgG isotype antibody was used as negative isotype control for the immunoprecipitation. For immunoprecipitation, the anti-IgG and anti-pSTAT3 antibodies were used (Cell Signaling Technology). We also included a nontemplate control in the PCR reaction (Appendix A). The following formula was used to determine the fold enrichment:∆CT (normalized IP) = Cp (IP) − Cp (IgG)
Fold Enrichement = 2 [−∆CT(normalized IP)]

### 2.13. Apoptosis Assays

WEHI-231 cells and immature B cells from 9-week-old mice were preincubated for 30 min with G129R (PRL-inh), GSK, or Stattic and incubated with PRL for 1 h before stimulating them with anti-IgM F(ab)2 (10 μg/mL) to induce clonal deletion or apoptosis for 48 h (WEHI-231) or 18 h (murine B cells). Cells were washed with PBS and incubated with Ghost-Red (Tonbo Biosciences, San Diego, CA, USA) at 4 °C for 30 min. For caspase-3 staining, the cells were permeabilized with Cytofix or Cytoperm (BD Biosciences) at 4 °C for 1 h, washed with Perm/wash, and incubated with anti-caspase-3-FITC at 4 °C for 1 h. Data were acquired using a MACSQuant Analyzer 10 cytometer and analyzed with FlowJo software. Cells cultured in a medium without inhibitors and anti-IgM F(ab)2 were used to compare PRL-induced apoptosis.

### 2.14. Statistical Analysis

The results were analyzed according to the distribution of the data (average and deviation). The Shapiro–Wilks normality test was used to determine the distribution of data. The quantitative independent variables were compared using the paired *t*-test. Differences between groups were determined using the ANOVA test. A value of *p* < 0.05 was considered significant; statistical analysis of the data was performed using SPSS Statistics 27 software (IBM, Armonk, NY, USA).

## 3. Results

### 3.1. Immature B Cells from Mice and WEHI-231 Cells Express the Long Isoform of the PRL Receptor

PRL can activate different cellular signaling pathways depending on the receptor isoform that is expressed. We determined the PRL receptor isoform expressed in BM B cells from the C57BL/6 control and lupus-prone MRL/lpr mice at 9 weeks of age. Here, we used bulk BM B cells and observed that only the long isoform was expressed (Figure 1A); the same was observed in WEHI-231 cells (Figure 1B). B cells in different stages of BM maturation were purified by sorting (Figure 1C), and we confirmed that only the long PRL receptor isoform was expressed throughout each stage of maturation in mice at 9 and 15 weeks of age. In C57BL/6 mice, we observed that the expression of the long isoform decreased as the B cell matured: 9-week-old mice, pro-B 0.050 ± 0.015, pre-B 0.044 ± 0.009, and immature 0.010 ± 0.004; 15-week-old mice, pro-B 0.057 ± 0.008, pre-B 0.029 ± 0.016, and immature 0.010 ± 0.003 (Figure 1D). In the MRL/lpr strain, we observed than the pro-B cells (0.012 ± 0.003) and immature B cells (0.011 ± 0.004) showed higher expression than pre-B cells (0.004 ± 0.004) (Figure 1E). Moreover, in 15-week-old MRL/lpr mice, in which elevated PRL levels have been documented [10,11], an increase in the expression of the long isoform was found in all populations that was greater in immature B cells (pro-B 0.067 ± 0.007, pre-B 0.061 ± 0.00, and immature 0.130 ± 0.024) (Figure 1E, Appendix A). In summary, only the long isoform of the PRL receptor was observed, and immature B cells were the stage of differentiation in which higher levels of expression were observed in lupus-prone MRL/lpr mice.

### 3.2. PRL Activates STAT3 in Immature B Cells from MRL/lpr Mice, Whereas in WEHI-231, It Activates the PI3K/AKT and STAT3 Signaling Pathways

We then determined the signaling components associated with the PRL receptor upon activation with recombinant PRL in immature B cells isolated from 9-week-old control and MRL/lpr mice. Since the JAK–STAT pathway is known to be activated by the long receptor isoform, we determined whether PRL activates the STAT kinases through STAT phosphorylation by flow cytometry. We found that PRL induced phosphorylation of STAT3 (pSTAT3) in MRL/lpr mice and confirmed this activity with an inhibitor of the PRL receptor (G129R) and a STAT3 inhibitor (Stattic). The pSTAT3 was measured by mean fluorescence intensity (MFI) and the percentage of positive cells (medium 30.12 ± 5.90 MFI, 5.98 ± 0.35%; PRL 42.85 ± 13.18 MFI, 10.45 ± 0.35%; G129R 30.13 ± 2.43 MFI, 5.53 ± 1.050%; Stattic 24.50 ± 2.73 MFI, 5.72 ± 0.45%) (Figure 2A–C). In addition, PRL activity was more prominent in MRL/lpr immature B cells, since in the lupus-prone strain only, the inhibitors significantly reduced pSTAT3 (Figure 2A). PRL did not activate STAT1, STAT5, AKT, or ERK phosphorylation in control or MRL/lpr mice. In agreement, no significant variations were observed in the activation of these kinases when we used the PRL inhibitor or Stattic (Appendix A).

Although PRL induced pAKT in WEHI-231 cells (medium 345.3 ± 6.56 MFI, 1.88 ± 0.40%; PRL 400.00 ± 41.80 MFI, 8.73 ± 0.94%; G129R 312.83 ± 29.68 MFI, 3.63 ± 0.50%; GSK, 247.4 ± 44.20 MFI, 3.10 ± 0.55%) (Figure 2D,E), a similar pattern of PRL activation of the signaling pathways was observed. PRL increased the levels of pSTAT3 (medium 155.0 ± 4.58 MFI, 16.23 ± 2.24%; PRL 174.0 ± 2.64 MFI, 26.17 ± 3.15%) that was reduced with other inhibitors (G129R 153.3 ± 7.23 MFI, 12.23 ± 1.66%; Stattic 148.0 ± 8.88 MFI, 9.23 ± 1.21%), as shown in Figure 2F,G, but no increase was observed in pSTAT1, pSTAT5, and pERK (Appendix A). These results support the hypothesis that PRL signals through STAT3 in immature B cells of MRL/lpr mice, which was confirmed in WEHI-231 cells, although in the latter, we also observed AKT activation.

### 3.3. The Pattern of PRL-Regulated Genes Correlates with Antiapoptotic Activity and Places STAT3 as a Central Transcription Factor

We have previously reported that PRL protects WEHI-231 cells from BCR cross-linking-induced apoptosis [12]. Here, we assessed whether PRL also alters apoptosis-related genes ex vivo in immature B cells isolated from MRL/lpr and C57BL/6 control mice. For this, we tested the gene expression array RT^2^ Profiler™ PCR Array Mouse apoptosis, which measures the expression of 84 genes related to death domain receptors, DNA damage and repair, extracellular apoptotic signals, and many other gene intermediaries and effectors of the apoptosis cascade. An unsupervised clustering based on genes with altered expression segregated mouse strains (Figure 3A). We proceeded to evaluate each strain independently, comparing the PRL condition against the unstimulated controls (Figure 3B,C). Upregulation of the antiapoptotic genes *Bcl2l2* (Bcl-w), *Bcl2a1a* (A1/Bfl-1), *Naip1* (Birc1), and *Birc5* (survivin) was observed in MRL/lpr immature B cells, and downregulation of many proapoptotic genes. A more divergent result was observed in C57BL/6 cells, with both antiapoptotic and proapoptotic genes upregulated and downregulated. However, *Birc5* was upregulated in both mouse strains (Figure 3D).

To assess the specific genes that were altered in MRL/lpr mice, we compared the PRL-altered genes against all additional conditions, such as unstimulated cells, plus the conditions of inhibition of the PRL receptor (G129R), and STAT3 (Stattic). An unsupervised clustergram showed a closer association between PRL and unstimulated cells than to the inhibitor conditions, probably suggesting the basal activity of the PRL receptor in MRL/lpr immature B cells (Figure 4A). A comparison of genes altered through PRL against all the other conditions showed upregulation of the antiapoptotic genes *Birc5* and *Bcl2a1a* and downregulation of proapoptotic genes *Casp8*, *Gadd45a*, and *Trp53bp2*, consistent with the protective role of self-reactive immature B cells (Figure 4B). A supervised clustergram also highlighted this set of genes, sharply separating PRL-activated cells from inactive cells (Figure 4C). All the genes that changed with the PRL stimuli are shown in Figure 4D. We confirmed the upregulation of *Birc5* by RT-PCR and that *Bcl2L1* (Bcl-xL) expression was not altered (Figure 4E).

We inferred the main transcription factors activated by PRL in MRL/lpr mice using the list of genes in Figure 4D; we observed that STAT3 was a central transcription factor regulating most of the genes in the list, four upregulated and five downregulated (Figure 5A–C). The transcription factors with the highest enrichment scores were STAT3, NFκB2, HIF1A, and E2F7 among the upregulated genes, and STAT3 and DDIT3 among the downregulated genes (Figure 5A). Protein–protein interaction networks were built using the transcription factors and targeted genes, for both the set of upregulated and downregulated genes independently (Figure 5B,C). This analysis confirmed the centrality of STAT3 and its direct influence on multiple genes, including *Birc5*, *Casp8*, and *Gadd45a*, three genes that were commonly altered by the control conditions (Figure 4B). A functional enrichment analysis using STRING placed the negative regulation of apoptosis as the second most important term of the upregulated protein–protein interaction network and the extrinsic apoptotic signal as the sixth term of the downregulated network (Figure 5B,C). STAT3 and *Birc5* appeared as part of the network that negatively regulated apoptosis.

### 3.4. pSTAT3 Binds to Promoters of the Bcl2 Family and Birc5 of Antiapoptotic Genes

We used the ChIP assay to determine the binding of pSTAT3 to the promoter sites of antiapoptotic genes that were central in this study. We chose *Bcl2l1*, a gene previously observed in our analysis of WEHI-231 cells [12], and also in C57BL/6 immature B cells (Figure 3D), and *Bcl2l2*, *Bcl2a1a*, and *Birc5*, some of the most central genes in the analysis of MRL/lpr cells (Appendix A). We corroborated the binding of pSTAT3 to the promoters of *Bcl2l1* (2.65 ± 0.21 fold enrichment) and *Birc5* genes (3.10 ± 0.28 fold enrichment) in PRL-stimulated immature B cells from C57BL/6 mice, whereas in MRL/lpr mice, pSTAT3 bound to the promoter sites of *Bcl2l2* (3.90 ± 0.28 fold enrichment), *Bcl2a1a* (6.05 ± 0.35 fold enrichment), and *Birc5* (6.20 ± 0.42 fold enrichment) genes (Figure 5D). In the WEHI-231 line, we observed that PRL induced the binding of pSTAT3 to the promoters of *Bcl2l1* (3.55 ± 0.35-fold enrichment) and *Bcl2l2* genes (3.15 ± 0.35-fold enrichment).

### 3.5. Prolactin Rescues Immature B Cells from Apoptosis via the STAT3 Pathway

To confirm that PRL signaling through STAT3 was responsible for rescuing immature B cells from apoptosis, we performed apoptosis assays (determined as the percentage of caspase-3 positive cells) in the presence of G129R (PRL-inh) as well as Stattic (STAT3 inhibitor). We observed that PRL rescued cells from apoptosis induced by BCR crosslinking with anti-IgM antibodies in MRL/lpr mice but not in C57BL/6 mice. When the PRL signal was inhibited, the effect of the hormone was reversed (MRL/lpr medium 19.17 ± 1.88; anti-IgM F(ab’)_2_ 47.26 ± 4.60; PRL 34.60 ± 0.82; G129R 51.90 ± 2.83; Stattic 52.07 ± 3.49) (Figure 6A). In C57BL/6 control mice, we observed a nonsignificant increase in apoptosis with the STAT3 inhibitor (Figure 6B). As we have previously reported [12], PRL rescued the WEHI-231 cells from apoptosis and the rescue was also abolished with the inhibitors (medium 31.17 ± 0.25; anti-IgM F(ab’)_2_ 84.40 ± 2.15; PRL 78.11 ± 0.32; G129R 83.87 ± 1.52; Stattic 83.5 ± 0.10) (Figure 6C).

## 4. Discussion

The role of PRL in apoptosis inhibition in cells of the immune system such as thymocytes [25] and T cells [26], and during B cell maturation in the BM [27], as well as in breast [26] and ovarian [28] cancer cells has been previously documented. Moreover, somatic gain-of-function STAT3 mutations are exclusive to T, NK, and B cell malignancies [29]. We have also documented the ability of PRL to rescue the immature B cells and WEHI-231 cell line from BCR cross-linking-induced apoptosis [12], which is an experimental scheme that mirrors the elimination of self-reactive clones and has been extensively studied as a model of B cell tolerance (clonal deletion). Nevertheless, the molecular mechanism through which PRL exerts antiapoptotic effects in these cells and how this hormone influences the development of SLE remain unclear. In this study, we have presented evidence for a novel mechanism, demonstrating, for the first time, that all stages of B cell maturation in the BM exclusively express the long isoform of the PRL receptor, both in lupus-susceptible and nonsusceptible mice. We also found that the PRL receptor expression increases with B cell maturation in lupus-susceptible mice, being the highest in immature B cells and coinciding with the time of the onset of the disease, which supports a mechanism of protection against the clonal deletion of autoreactive clones. Protection against the apoptosis of cancer cells is also mediated by the long isoform of the PRL receptor [28].

Our data support that PRL prevents immature B cell clonal deletion through STAT3 signaling, since its inhibition affected PRL functional activity and STAT3 was the most central transcription factor in the bioinformatic analysis. We did not find evidence of AKT or ERK activation, as these kinases were not phosphorylated upon PRL treatment. Furthermore, ERK was also included in the set of genes analyzed by the array and its expression was not altered. In contrast, we found that in WEHI-231 cells, PRL signals through STAT3 and PI3K–AKT. PRL signaling through multiple pathways has been described for other cell lines, such as PI3K–AKT and STAT in Nb2 rat lymphoma cells [30,31], and STAT5, AKT, and ERK1/2 in T47D breast cancer cells [32]. Differential PRL pathway activation may be influenced by the isoform of the receptor expressed and by the tyrosine residue(s) phosphorylated upon receptor activation. The intracellular domain (IC) of the PRL receptor long isoform contains several tyrosine residues. The phosphorylation of IC tyrosines creates binding sites for SH2 domain-containing signaling proteins. The specificity of SH2 proteins binding to phospho-tyrosines is influenced by the surrounding amino acid sequences [33].

Further support for the antiapoptotic effect of PRL through STAT3 was found through the use of the STAT3 inhibitor Stattic. Both Stattic and PRL-inh, which is a prolactin-based peptide containing a G129R mutation that acts as an antagonist of the PRL receptor, increased the frequency of apoptotic caspase-3 positive immature B cells. Remarkably, PRL seems to solely counteract negative selection in the SLE-susceptible MRL/lpr mice. Although immature B cells from C57BL/6 mice also express the long PRL receptor isoform, PRL could not rescue them from BCR cross-linking-induced apoptosis. This may have different possible explanations: while the protein expression of the PRL receptor decreased in immature B cells from C57BL/6 mice, it increased and was higher in MRL/lpr mice (Appendix A) [11], rendering PRL signaling insufficient to counteract death signals in the former. This has been observed in humans, where T cells from patients with SLE express higher levels of the PRL receptor than those from healthy subjects; upon activation with anti-CD3/CD28 plus PRL, increased IFNγ secretion is only observed in the T cells of patients with SLE [34]. Similarly, PRL-induced signaling was only observed in colon cancer cells and cell lines with a higher PRL receptor expression than in normal colonic epithelial cells [35].

PRL seems to counteract clonal deletion signals through a complex pattern of transcription in which there is a synchronized upregulation of several antiapoptotic genes and downregulation of proapoptotic genes. The transcriptional and gene network analyses identified *Birc5*, *Bcl2a1a*, *Bcl2l2*, *Xiap*, *Igf1r*, *Trp53bp2*, and *Naip1* among the antiapoptotic genes whose expression was significantly enhanced, and *Casp8* and *Gadd45a* among the proapoptotic genes lost. Notably, although the pattern of genes altered in C57BL/6 cells is also very complex, there is no clear-cut final outcome, since multiple proapoptotic genes exhibited increased expression and multiple antiapoptotic genes were lost upon PRL activation. *Bcl2a1a*, *Bcl2l2*, and *Trp53bp2* are among these discordant genes, which are important for rescuing MRL/lpr immature B cells. Therefore, different levels of expression of the PRL receptor at the protein level and a discordant network of the regulation of genes related to apoptosis may explain, on one hand, why we did not observe STAT3 phosphorylation and rescue of apoptosis in C57BL/6 immature B cells, and, on the other hand, the capacity of PRL to trigger SLE specifically in individuals with genetic susceptibility.

The experimental and bioinformatics analyses identified STAT3 as the most central transcription factor driving the expression of apoptosis-related genes, but HIF1α, NFkB2, E2F7, and HES1 also appeared to participate. The functions of STAT3 in vivo have been extensively studied in both health and disease, particularly in cancer. Most STAT3 target genes are key participants of cell growth, cell cycle progression, survival, transformation, invasion, and metastasis [36,37,38]. STAT3 activation is also required for the differentiation of B cells into plasma cells [39] and for B cell maturation in BM [27]. The ChIP analysis also confirmed the link between PRL and STAT3, with *Bcl2a1a*, *Bcl2l2*, and *Birc5*. STAT3 activation of *Birc5* has been reported in paclitaxel- and cisplatin-resistant ovarian cancer cells, and the abrogation of STAT3 signaling results in a reduced expression of *Birc5*, circumventing chemoresistance [40,41]. *Birc5* is also downstream of STAT3 transcription activity in triple-negative breast cancer, and STAT3 binding to the *Birc5* promoter has been demonstrated through ChIP analysis [42,43]. STAT3 also induces chemoresistance in diffuse large B-cell lymphomas through a decreased expression of *Casp8*, among many other genes [44]. STAT3 downregulation of *Casp8* has also been shown in the human leukemia HL-60 cell line [45]. Another neoplasm in which STAT3 has been linked to *Birc5* is acute myeloid leukemia (AML). Common mutations in the kinase domain of Flt3 result in the constitutive phosphorylation of STAT3 and increased levels of *Birc5* expression, which protects AML cells from apoptosis [46]. All these studies support the association between STAT3 and resistance to apoptosis, and confirm an important role for *Birc5* in hematological neoplasms and solid cancers.

An extensive body of evidence links STAT3 with autoimmune diseases. Most of the evidence is related to the capacity of STAT3 to influence the differentiation of lymphoid cells, such as Th17 and Treg CD4 T cells [47]. Indeed, gain-of-function mutations have been reported in patients with autoimmune conditions [48,49]. A different mechanism may operate in STAT3’s relationship with SLE, since Stattic inhibition of STAT3 blocked the secretion of autoantibodies from B cells isolated from patients with SLE [50]. Stattics have also been used to delay the onset of disease in MRL/lpr mice, reducing the levels of clinical hallmarks of SLE, such as nephritis, renal and skin lesions, proteinuria, and serum autoantibodies [51,52]. Moreover, inhibitors of JAK2, which is a kinase upstream of STAT3, are currently in clinical trials to treat SLE [53], as well as open clinical trials (NCT03616912 [Last Update 20/01/2021], NCT03616964 [Last Update 19/10/2020]).

Less is known about the contribution of STAT3-induced cell survival for autoimmune diseases and whether Bcl2 and inhibitor of apoptosis (IAP) family members play a role, as has been documented for cancer cells. We have previously documented increased *Birc5* expression in pro-B and immature B cells in an SLE-prone mouse strain, which is associated with high levels of PRL [11]. There is also evidence of *Bcl2a1a*, *Bcl2l2*, and *Birc5* expression in bone marrow B cells. For instance, *Bcl2l2* and *Birc5* expression was observed throughout the small pre-B cell stage but it was lost in immature B cells in mice without an autoimmune disease genetic background [54]. Mice overexpressing *Bcl2a1a* in the hematologic compartment develop an aggressive malignant disease characterized as leukemia or lymphoma of B cell origin [55]. *Bcl2l2* also plays a critical role in B cell survival and lymphomagenesis [56]. In contrast, *Birc5* is overexpressed in numerous malignant diseases, as well as in autoimmune diseases, including multiple sclerosis and myasthenia gravis [57,58].

Collectively, our data suggest that PRL protects autoreactive immature B cells from BCR cross-linking-induced apoptosis through STAT3 and a complex network of apoptosis-related genes (see Figure 7 for a current working model). Increased PRL levels have been associated with disease activity, and the increased immune complex deposition is a hallmark of the damage to tissues and organs that characterizes SLE. We have previously documented that elevated levels of PRL increase the absolute number of splenic transitional-1 B cells in MRL/lpr mice [10]. In future studies, it will be important to follow the influence of PRL in the maturation of B cells in secondary lymphoid organs, particularly to understand if PRL facilitates the activation of self-reactive clones and their differentiation into plasma cells, in order to better understand the effect of PRL on the pathogenesis of SLE.

## 5. Conclusions

All B cell maturation stages in bone marrow express the prolactin receptor long isoform in both wild-type and MRL/lpr mice, but its expression increased only in the immature B cells of the latter, particularly at the onset of lupus. In these cells, activation of the PRL receptor promoted the phosphorylation of the transcription factor STAT3, demonstrating that it binds to the promoter of the anti-apoptotic genes *Birc5*, *Bcl2a1a*, and *Bcl2l2*, and confirming that the PRL receptor signaling pathway rescues immature B cells from apoptosis-induced BCR aggregation.

## 6. Limitations of Study

A limitation of the present study is that we compared the C57BL/6 mouse strain with the MRL/lpr lupus prone strain. The MRL/lpr strain carries a mixed genetic background plus a mutation in the *Fas* gene. Since the lupus-susceptible MRL-MpJ strain is the closest genetically to MRL/lpr, future experiments should examine PRL’s anti-apoptotic function in this strain to assess the influence of Fas on the prolactin–STAT3 axis. Furthermore, although we specifically wanted to assess apoptosis in this study, RNA sequencing would help to unveil the contribution of other mechanisms to the activity of PRL in immature B cells.

## Figures and Tables

**Figure 1 cells-10-00316-f001:**
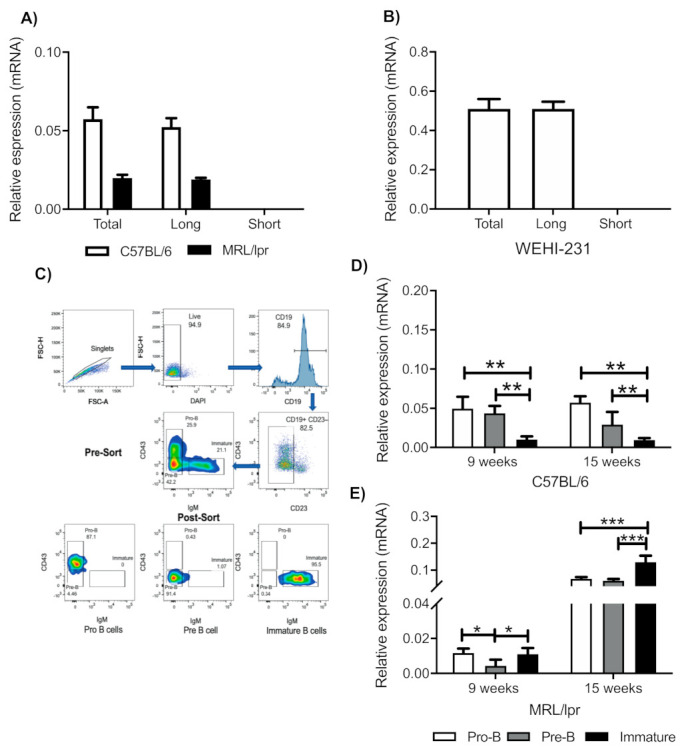
Relative expression of the long prolactin (PRL) receptor isoform. Bulk bone marrow (BM) B cells, and pro-B, pre-B, and immature-B cells were purified through flow cytometry and subjected to real-time (RT) PCR to determine the relative expression and identity of the PRL receptor isoforms. (**A**) Expression of the long and short isoforms in bulk BM B cells from C57BL/6 and MRL/lpr mice (the murine breast cancer cell line EpH4 1424 was used as positive control for expression of the long and short PRL-receptor isoforms), and (**B**) in the WEHI-231 cell line. (**C**) Demonstration of the gating strategy for sorting. Doublets were excluded by gating on FSC-H × FSC-A, and live cells were gated in the DAPI negative, and with the CD23 negative gate we excluded all mature recirculating B cells; Pro-B cells (CD43+IgM–), Pre-B cells (CD43–IgM−) and immature B cells (CD43–IgM+) were sorted. (**D**) Expression of the PRL isoforms in pro-B, pre-B, and immature B cells from C57BL/6 mice at 9 and 15 weeks of age and (**E**) in MRL/lpr mice. Three independent experiments were conducted and a pool of three mice was used in each experiment. Pooled data are presented as mean ± SD. *, *p* < 0.05; **, *p* < 0.01; and ***, *p* < 0.005 using one-way analysis of variance (ANOVA).

**Figure 2 cells-10-00316-f002:**
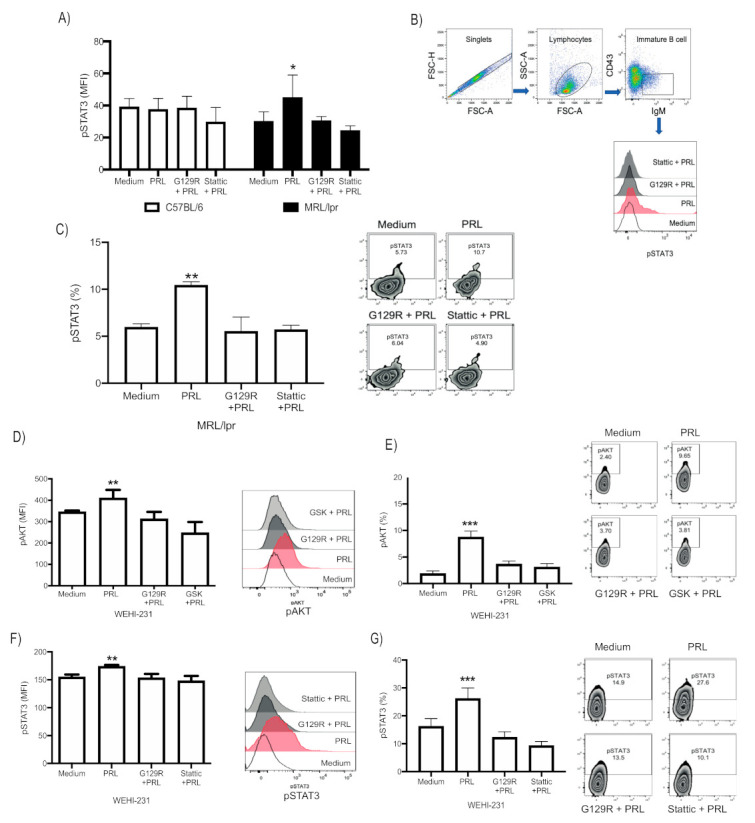
Analysis of the signaling pathways activated by PRL in immature B cells. (**A**) BM B220+CD23− cells purified by MACS Cell Separation from 9-week-old C57BL/6 and MRL/lpr mice were preincubated for 30 min with the following inhibitors of: PRL receptor (G129R) and STAT3 (Stattic). The cells were then incubated for 30 min with PRL and stainedwith anti-CD43, anti-IgM, and anti-pSTAT3 to subsequently determine the median fluorescence intensity (MFI) of phosphorylation of pSTAT3 in immature B cells (CD43−IgM+). Four independent experiments were performed, and each experiment was conducted in triplicate. Pooled data are presented as means ± standard deviation (SD). * *p* < 0.05 using ANOVA. (**B**) Demonstration of the gating strategy for the flow cytometric analysis of pSTAT3 in immature B cells. Doublets were excluded by gating on FSC-H × FSC-A and lymphocytes were identified by their scatter properties (FSC-A × SSC-A plot). The surface CD43−IgM+ population represents immature B cells. The histogram shows one experiment representative of pSTAT3 in MRL/lpr immature B cells. (**C**) Percentage of STAT3 positive immature B cells in MRL/lpr mice. Pooled data are presented as means ± SD; **, *p* < 0.01 using ANOVA. A zebra plot of one representative pSTAT3 experiment is shown (right). (**D**,**E**) WEHI-231 cells were preincubated for 30 min with G129R, Stattic, and GSK inhibitors and then incubated 30 min with PRL to determine the phosphorylation of AKT. (**D**) MFI graph and a histogram from one representative experiment; (**E**) graph of percentages of positive cells and a zebra plot. (**F**,**G**) Phosphorylation of STAT3 shown as an MFI graph and histogram (**F**), and graph of percentages and a zebra plot (**G**). Four different experiments were performed, each in triplicate. Pooled data are presented as means ± SD. **, *p* <0.01 and ***, *p* < 0.005 using ANOVA.

**Figure 3 cells-10-00316-f003:**
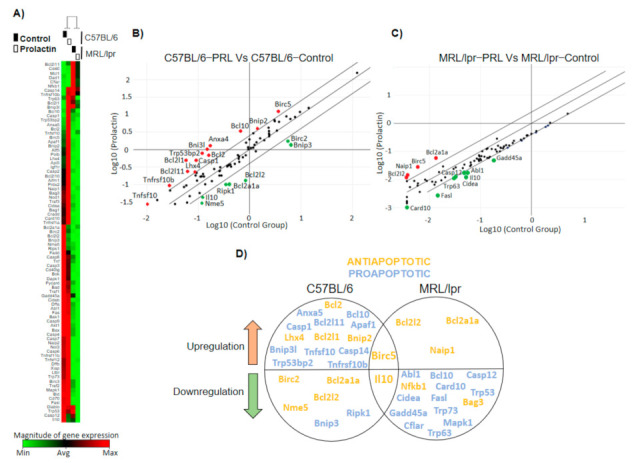
Apoptosis-related genes commonly regulated by PRL in MRL/lpr and C57BL/6 mice. Immature B cells from 9-week-old C57BL/6 and MRL/lpr mice were incubated for 30 min with or without PRL. After stimulation, the expression of 84 genes related to apoptosis was determined. (**A**) Unsupervised heat map and clustergram of genes transcriptionally altered by the experimental conditions. Scatter plot of genes differentially expressed after PRL stimulation in immature B cells from (**B**) C57BL/6 and (**C**) MRL/lpr mice. Red and green dots in (**B**,**C**) represent upregulated and downregulated genes, respectively. (**D**) Venn diagram representing differentially expressed genes shared by both mouse strains.

**Figure 4 cells-10-00316-f004:**
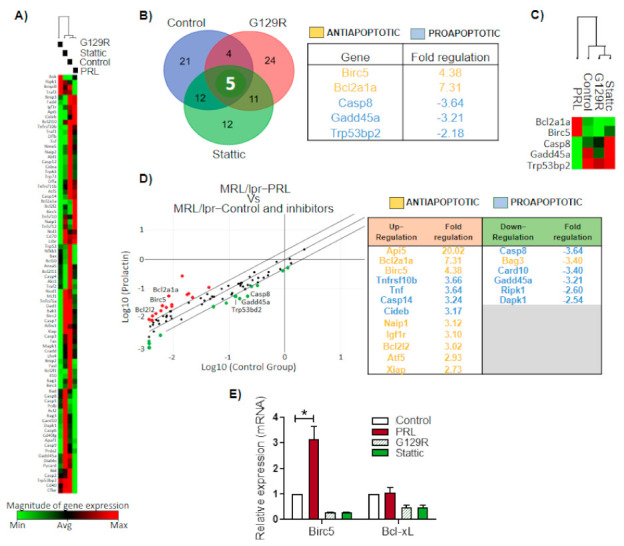
PRL regulation of apoptosis-related genes in MRL/lpr mice. Immature B cells from 9-week-old MRL/lpr mice were preincubated for 30 min with G129R or Stattic and stimulated for 30 min with PRL. Expression of apoptosis-related genes was determined in the PRL condition vs. the unstimulated and inhibitor conditions. (**A**) Unsupervised heat map and clustergram of genes transcriptionally altered by the experimental conditions. (**B**) Venn diagram and (**C**) heat map representing genes shared by control, G129R, and Stattic conditions and differentially expressed with respect to the PRL stimulation condition. (**D**) Scatter plot of genes differentially expressed in supervised analysis of PRL stimulation compared with control, G129R, and Stattic conditions. Red and green dots represent upregulated and downregulated genes, respectively. (**E**) Validation of PRL-induced *Birc5* and *Bcl2l1* (Bcl-xL) upregulation using RT-PCR. Three independent experiments were performed and a pool of two mice was used in each experiment. Pooled data are presented as means ± SD. *, *p* < 0.05 using ANOVA. In (**B**,**D**), the identity, fold change, and apoptotic function of the altered genes are shown in tables.

**Figure 5 cells-10-00316-f005:**
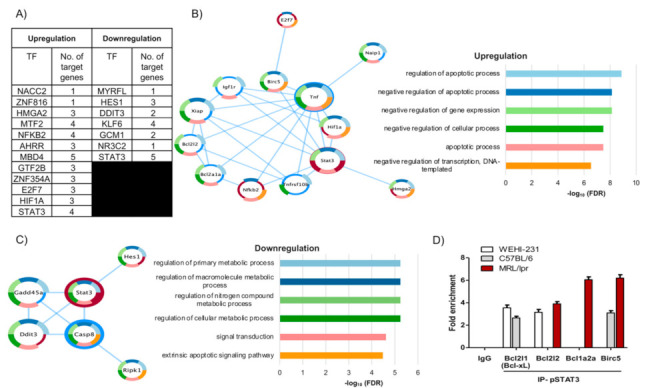
Bioinformatics and chromatin immunoprecipitation analyses of the transcriptional regulation of the apoptosis resistance genes. (**A**) Predicted transcription factor enrichment based on differentially expressed genes upon PRL activation of MRL/lpr immature B cells from 9-week-old mice. Protein–protein interaction networks generated using the set of transcription factors and the related upregulated genes (**B**) or the set of transcription factors and the related downregulated genes (**C**) as input. The size and border width of the nodes represent closeness centrality and betweenness centrality, respectively; the nodes with a red border are central transcription factors, nodes with blue borders are the direct genes under the transcription factor regulation, and donut charts represent gene ontology processes enriched in the networks. (**D**) pSTAT3 ChIP of immature B cells from 9-week-old C57BL/6 and MRL/lpr mice and WEHI-231 cells after stimulation for 1 h with PRL. The plot represents the fold enrichment at different antiapoptotic gene promoter sites.

**Figure 6 cells-10-00316-f006:**
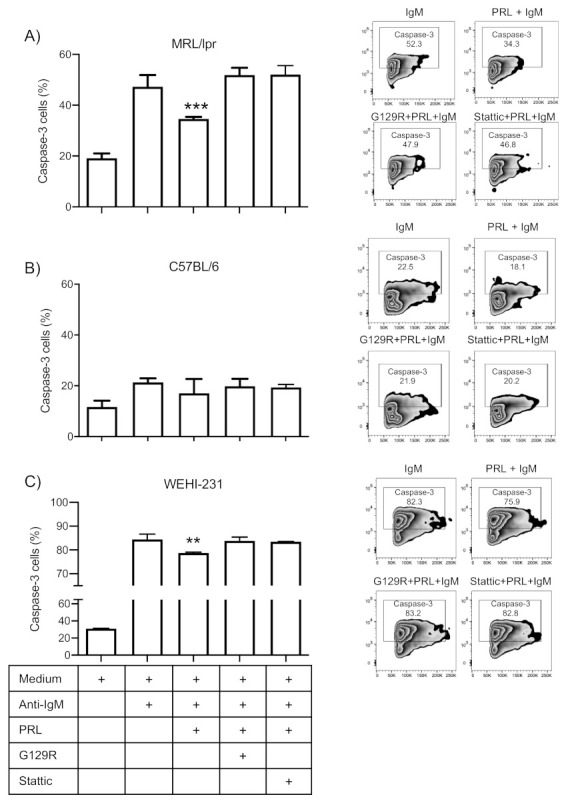
Effect of PRL receptor signaling on immature B cell apoptosis. Purified immature B cells from 9-week-old MRL/lpr (**A**) and C57BL/6 mice (**B**), and WEHI-231 cells (**C**) were activated for 1 h with PRL after either PRL receptor (G129R) or STAT3 (Stattic) signaling inhibition (30 min), and then incubated with anti-IgM antibody to induce BCR cross-linking-mediated apoptosis. Zebra plots represent the percentage of caspase-3-positive apoptotic cells. Four different experiments were performed in triplicate. Pooled data are presented as means ± SD. **, *p* < 0.01, ***, *p* < 0.005; using ANOVA.

**Figure 7 cells-10-00316-f007:**
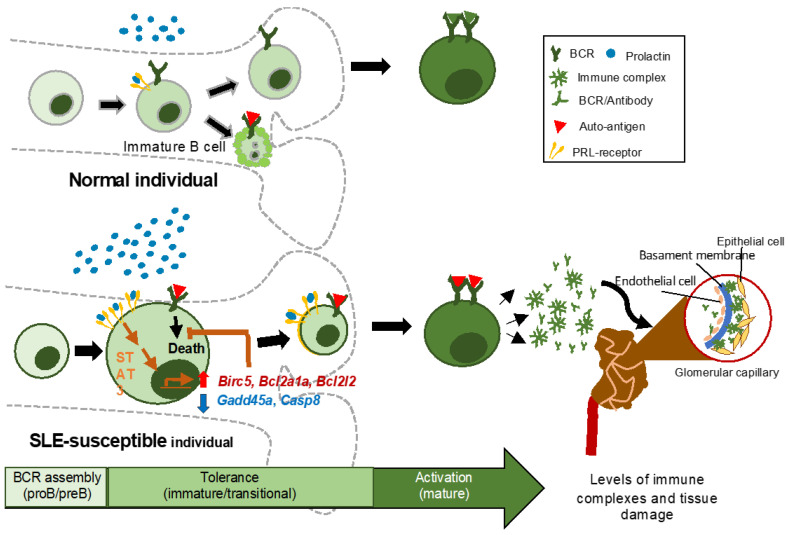
Working model of the mechanism of action of PRL in immature B cells from mice that developed SLE. Immature B cells from SLE-susceptible mice display increased levels of PRL receptor expression restrictive to the long isoform, together with higher levels of serum PRL. This combination results in the heightened activation of STAT3 and altered expression of genes related to the control of apoptosis, such as *Birc5*, *Bcl2a1a*, *Bcl2l2*, *Gadd45a*, and *Casp8*, and also of other Bcl2 and inhibitor of apoptosis (IAP) family members, which increase the survival of immature B cells, including those cells undergoing clonal deletion because of rearranged self-reactive antigen receptors. By allowing these autoreactive clones to continue their maturation process and eventually become plasma B cells, PRL contributes to the increased levels of autoantibody and tissue damage due to immune complex deposition that characterize SLE.

## Data Availability

The data presented in this study are available on request from the corresponding author.

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
