# Peer review of "Prolactin Rescues Immature B Cells from Apoptosis-Induced BCR-Aggregation through STAT3, Bcl2a1a, Bcl2l2, and Birc5 in Lupus-Prone MRL/lpr Mice"

_cells, 2021, doi:10.3390/cells10020316_

Round 1

Reviewer 1 Report

The manuscript by Flores-Fernandez examines the rescue of immature B cells by the neuroendocrine hormone prolactin (PRL) in lupus prone MRP/lpr. The manuscript would of general interest in that the investigator (largely) show that PRL inhibits apoptosis of immature B cells through a Stat3 driven mechanism. The conclusion while of interest are limited by the abbreviated examination of the transcriptome by a limited 84 gene set panel, when RNA-seq should have been performed. In a sense, the conclusions generated by the manuscript seems "forced" by the nature of its design. Specific concerns are listed below:

1) Inadequate description of PRL-inh and Static are provided. No presentation of what the off-target effects of these agents is provided. Appropriate knock-down studies should be presented.

2) Why did the authors pick G129R PRL as a PRL antagonist? It is not, instead it is a partial antagonist/partial agonist. If the authors wanted to use of PRL antagonist then they should have used del1-9 G129R PRL.

3) The antibodies used in the ChIP experiments have not been validated for ChIP.

4) Fig. 1. An independent method for PRLr determination needs to be also presented, whether that be by IP or Flow.

5) Fig. 5 Fundamental quality control of the ChIP is not presented.

6) Fig. S1. If the author's PRL antagonist are working appropriately there should be a significant reduction in activated Stat5. This was not seen and the investigators need to explain.

Reviewer 2 Report

Recently neuro-immune interaction is attracting great interest.

The authors have been studying roles for prolactin in abnormal B cell maturation in a SLE model MRL/lpr and in protection against apoptosis induced by cross-linking of BCR on WEHI231. Based on these studies, they have tried to clarify the molecular mechanisms of prolactin signals in pathophysiology of autoimmunity in MRL/lpr. 

The methods using sorting, PCR-Array, RT-PCR, flow cytometry, and chromatin immunoprecipitation are appropriate. But using C57BL/6 as a control for MRL/lpr is not appropriate especially in PCR-Array study, because genetic backgrounds may affect the differential gene expression between two mice lines.    

Additional experiments confirming expression of important genes between MRL/lpr and MRL/MpJ will rescue the data obtained with microarray comparing   MRL/lpr vs C57BL/6. 

Anti-apoptostic roles for prolactin-dependent STAT3 activation is clearly demonstrated by flow-cytometrical detection of pSTAT3 and apoptosis, which were confirmed with specific inhibitors for PRL receptor or STAT3. Furthermore, these findings are supported by the results of chromatin immunoprecipitation and bioinformatic analyses. Thus, overall data are reliable.

If the effects of genetic background in this study are excluded from current results, this paper could demonstrate the critical roles for prolactin-STAT3 axis in the molecular mechanisms of autoimmunity as well as hyper-activation of B cells in a murine SLE model.    

Minor points

Line 113;APE-conjugated anti-STAT1 → APC- or PE- ?

Round 2

Reviewer 2 Report

Although the author has listed 5 papers(see below; strains used in each paper are summarized.), no paper can show validity of comparison between MRL/lpr and C57BL/6 alone. Paper 2) and 5) includes MRL+/+(MRL-MpJ) as a control in comparison of MRL/lpr and C57BL/6, which is appropriate as I suggested. If the authors are interested in the effects of genetic backgrounds but not a single Fas mutation, a third party strain should be added as paper 1) and 3).

1)Int. Immunol 2002, 14:963    Ig-Tg in B10.A, C57BL/6, and MRL-lpr/lpr        

2)Gene Immun 2006, 7:156      MRL/lpr, MRL+/+, and C57BL/6                     

3)J Immunol 2006, 177:1120 MRL/lpr TdT-/-, C57BL/6 TdT-/-, BALB/c and NZB

4)Blood 2007, 110:1595 (Study of AML;there is no comparison of B6 vs MRL/lpr)                                                                                                

5)J Immunol 2008; 180: 5670.   C57BL/6, Mrl-MpJ, and Mrl Fcgr2b-/- 

Thus the comparison utilized in this paper is not perfect. Since MRL-MpJ itself is also autoimmune-susceptible strain, comparison of lupus prone MRL-MpJ vs normal C57BL/6 also has a significant value.  But on the other hand, even the comparison of MRL/lpr and C57BL/6 has resulted in presentation of substantial novelty and valuable information because of systematic and sound methods utilized in this paper. Furthermore, pandemic of SARS-CoV-2 has brought us very difficult situation not only in medical but also academic areas. I would like to suggest addition of such a sentence in discussion as "Causative roles for Fas mutation or the genetic background of MRL other than Fas mutation in this mechanism deserves further analyses in the future." or "examining operation of the prolactin-STAT3 axis in the lupus susceptible MRL-MpJ background without Fas mutation is next issue.".  

Minor points

1.line-445  WEHI-213 → WEHI-231                                                        

2.coverletter  (C67BL/6). → (C57BL/6).

Author Response

See attached please
